# COVID-19 and Pneumonia detection and web deployment from CT scan and X-ray images using deep learning

Nahid Islam, Abu S. M. Mohsin ⓘ *, Shadab Hafiz Choudhury ⓘ, Tazwar Prodhan Shaer, Md. Adnan Islam, Omar Sadat, Nahid Hossain Taz

Department of Electrical and Electronics Engineering, Nanotechnology, IoT and Machine Learning Research Group, BRAC University, Dhaka, Bangladesh

* asm.mohsin@bracu.ac.bd

## Abstract

During the COVID-19 pandemic, pneumonia was the leading cause of respiratory failure and death. In addition to SARS-COV-2, it can be caused by several other bacterial and viral agents. Even today, variants of SARS-COV-2 are endemic and COVID-19 cases are common in many places. The symptoms of COVID-19 are highly diverse and robust, ranging from invisible to severe respiratory failure. Current detection methods for the disease are time-consuming and expensive with low accuracy and precision. To address such situations, we have designed a framework for COVID-19 and Pneumonia detection using multiple deep learning algorithms further accompanied by a deployment scheme. In this study, we have utilized four prominent deep learning models, which are VGG-19, ResNet-50, Inception V3 and Xception, on two separate datasets of CT scan and X-ray images (COVID/ Non-COVID) to identify the best models for the detection of COVID-19. We achieved accuracies ranging from 86% to 99% depending on the model and dataset. To further validate our findings, we have applied the four distinct models on two more supplementary datasets of X-ray images of bacterial pneumonia and viral pneumonia. Additionally, we have implemented a flask app to visualize the outcome of our framework to show the identified COVID and Non-COVID images. The findings of this study will be helpful to develop an AI-driven automated tool for the cost effective and faster detection and better management of COVID-19 patients.

## A. Introduction

Novel Coronavirus disease 2019 (COVID-19) first emerged as a global pandemic in December 2019, when the first outbreak was reported in Wuhan, Hubei province, China [1]. COVID-19 is caused by Severe Acute Respiratory Syndrome Coronavirus 2 (SARS-CoV-2), whose highly transmittable strength, robust adaptability and exponential growth rate have puzzled antiviral drugs or vaccines available for treatment [2]. The WHO declared COVID-19 as a global pandemic on March 11, 2020, and announced 188,332,972 confirmed cases and reported 4,063,453 deaths globally as of July 16, 2021. It remains a major problem even in 2024, with over 1.1 million cases reported in December 2023 [3].

**Data Availability Statement:** All relevant data are within the paper and its Supporting Information files.

**Funding:** The author(s) received no specific funding for this work.

**Competing interests:** The authors have declared that no competing interests exist.

In general, COVID-19 patients show syndromes ranging from pneumonia, dry cough, fever, muscle spasms, along with acute respiratory distress syndrome (ARDS). In many cases, COVID-19 was hard to distinguish from pneumonia, and while treatments for the latter were effective they could not alleviate all symptoms or prevent transmission. Due to its high transmissibility, diagnosing and limiting the accelerated spread of COVID-19 is a high priority [4]. In order to prevent the transmission and detect the positive cases immediately, the necessity for supplemental diagnostic tools has increased since there are few accurate automated toolkits available for COVID-19 early detection. Early detection of COVID-19 is necessary and critical as it can allow us to prevent the transmission and further help us to take adequate treatment procedures towards the patients. Reverse-transcription polymerase chain reaction (RT-PCR) is broadly acknowledged as the gold standard for COVID-19 diagnosis yet this methodology experiences low sensitivity in the early stages, which may further prolong the transmission. This method transforms RNA to DNA and follows up with PCR to analyze the DNA, which later can be used to detect SARS-CoV-2 [5]. In general, this process takes 4–6 hours to return results. However, RT-PCR is that it gives inaccurate results approximately 5% of the time. Also, this test has other disadvantages such as the time taken to give results, the need for trained professionals, high incidents of false-positive results, and in many places shortage of kits. In recent research, CT chest images of the lungs and soft tissues have been studied for COVID-19 detection. The usage of CT is often restricted due to high radiation doses and expenses. Chest X-Ray is a radiological inspection with a non-invasive low radiation dose that can effectively image the lungs both time and cost-efficiently [6]. Chest radiography (X-Ray) is one of the most prominent techniques applied for the comprehensive analysis of pneumonia. Chest X-Rays can emerge as an adequate approach for the early detection of COVID-19 though radiographic feature similarities between COVID-19 and pneumonia may lead to challenging scenarios for radiologists.

In order to get precise and accurate detection of COVID-19, the coupling of Artificial Intelligence (AI) with radiological imaging has emerged as a prominent solution [7,8]. In this investigation, a comprehensive analysis of multiple models for automatic COVID-19 detection utilizing raw chest X-Ray images is performed. We introduced an effective scheme employing relevant available X-Ray images to train deep neural networks so that the trained models can be efficiently deployed to identify COVID-19 cases even with comparatively few COVID-19 X-Ray images available. On the other hand, X-ray images and CT Scan images are a widespread methodology of detection of a number of diseases. In places where there are thousands of affected people with insufficient testing kits and a lack of trained professionals, these methods of detecting COVID-19 with X-ray and CT scan images are effective. Densely populated regions can use COVID-19 detection using X-ray and CT scan images which will allow early detection of COVID-19. Though the limited availability of the images of affected and non-affected samples made the training of deep learning algorithms inefficient previously, with the advance of AI, we can now examine the performance of our model using limited resources. This research has used deep learning algorithms to train our program to detect COVID and Non-COVID patients by entering their X-ray and CT scan images. We applied VGG-19, ResNet-50, Inception V3 and Xception algorithms to train our framework and obtained high test accuracies ranging from 86% to 99%.

## B. Literature review

Deep learning performs a vital role in the biomedical image classification area due to its unique feature extraction and pattern recognition ability. In order to detect COVID-19 from CT scans and X-ray images, a convolutional neural network (CNN) has been the preferred approach.

Apostolopoulos et al. applied an evolutionary neural network for robust differentiation and automatic detection of COVID-19 with both typical pneumonias and COVID-19-induced pneumonia [9]. Chen et al. developed a multilayer deep learning-based model that has successfully detected COVID-19 with high sensitivity in a shorter time from chest X-ray images [10]. Khan et al. proposed a framework employing the deep learning architecture for COVID-19 detection using normal, bacterial and viral pneumonia cases [11]. Sahinbas and Catak presented a comparative analysis of performance measures using VGG16, VGG19, ResNet, DenseNet and InceptionV3 models on chest X-ray images [12].

Punn and Agarwal exhibited great sensitivity and precision on X-ray images using ResNet, InceptionV3, Inception-ResNet models for COVID-19 detection [13]. Hemdan et al. successfully used advanced deep learning models for COVID-19 diagnosis in X-ray images and proposed a highly efficient COVIDX-Net model containing seven individual CNN models [14]. Maghdid et al. presented a deep neural network-based method coupled with a transfer learning strategy for automatic detection of COVID-19 pneumonia [15]. Ghoshal and Tucker studied the uncertainty in deep learning solutions for COVID-19 detection in X-ray images using Drop Weights based Bayesian Convolutional Neural Networks (BCNN) [16]. Farooq and Hafeez suggested COVID-ResNet, an efficient fine-tuned and pre-trained ResNet-50 architecture for the early detection of COVID-19 pneumonia screening [17]. The COVID-ResNet framework achieved an accuracy of 96.23% on a multi-class classification on COVID-19 infection dataset. Chen et al. proposed an advanced Residual Attention U-Net for the automated multi-class segmentation procedure on COVID-19-related pneumonia applying CT images [18]. Narin et al. employed ResNet-50, InceptionV3 and Inception-ResNetV2 for 2-class classification and achieved the best accuracy of 98% with a pre-trained ResNet-50 model [19]. The following sections discuss the detailed structure of the proposed CAD scheme in this paper and the impact of the algorithms on COVID-19 detection.

The majority of these works achieved good results on either X-ray images or CT scans individually, but do not show a holistic comparison between the two. It is important to know which modalities perform best with which deep learning architecture. Additionally, we aimed to test certain models that have not been tested before, such as Xception.

## C. Datasets

**CT scan images:** The CT scan images dataset we used to feed our algorithm is the public covid-chestxray-dataset dataset, stored in a GitHub repository [20]. During the first of January 2021, the dataset consisted of 746 unique CT image scans of patients. The dataset contains meta information of patient's details and any other diseases the patient may have.etc. COVID19-related papers from medRxiv and bioRxiv etc. [21,22] were also used for gathering images for the work. It is also mentioned that this dataset is constantly updated with new images of patients. More details of the dataset can be found at [20]. Out of the 746 images, there were 349 COVID-19 affected images and 397 non-COVID-19 images. Fig 1 shows some samples from the dataset containing positive and negative Covid19 CT scan images.

**X-ray images:** We have collected the X-ray images for our models from two different datasets. We have collected COVID-19 infected images from a publicly open GitHub repository [23]. It consisted of chest X-ray images of patients which are positive or suspected of COVID-19 or MERS, SARS, ARDS. Some of the data are collected indirectly from hospitals and physicians.

The second dataset we used for X-ray images is from a public Kaggle dataset [24] from which we have collected images of normal, bacterial pneumonia and viral pneumonia infected patients. Although it contains images of COVID19 infected patients, we have only used the negative images because there was insufficient number of covid-19 images.

Positive COVID-19 CT Scan        Negative COVID-19 CT Scan

**Fig 1. COVID19 CT scan sample.**

Overall, we used 930 X-ray images of Covid-19 infected patients, 880 Normal images, 650 Bacterial Pneumonia and 412 Viral Pneumonia images. Fig 2 shows some samples of positive and negative Covid-19 Xray images.

## D. Deep learning models and methods

### The chosen deep learning models are described in this section

**Inception v3:** Inception V3 CNN based deep neural network with 48 hidden layers of module and primarily contains 13x3 and 5x5 convolutions [25]. Inception v3, trained on ImageNet [26], could classify images into a total of 1000 categories, including keyboard, pencil, mouse, and many other animals. The model we used was pretrained on more than one million images from the ImageNet dataset [27]. The original layers were frozen and several additional layers were added for fine-tuning. These are shown in Fig 3.

**VGG19:** VGG-16 has 16 convolutional and 3 fully connected layers. It used ReLU as the activation function, just like in AlexNet. VGG-16 had 138 million parameters. A deeper version, VGG-19, was also constructed along with VGG-16. The Architecture is shown in Fig 4 [28].

Positive COVID-19 XRay        Negative COVID-19 XRay

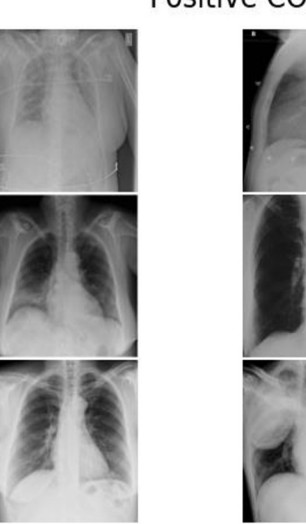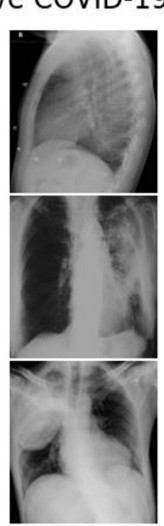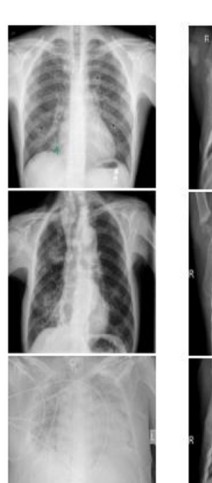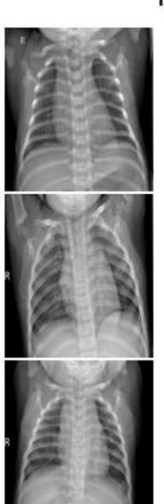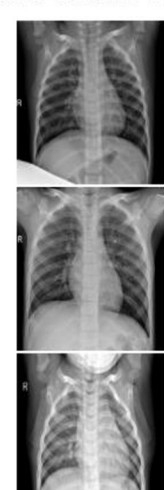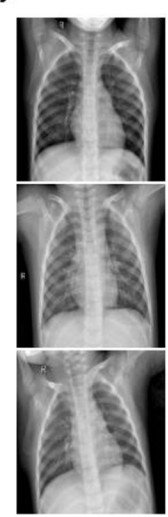

**Fig 2. COVID19 chest Xray sample.**

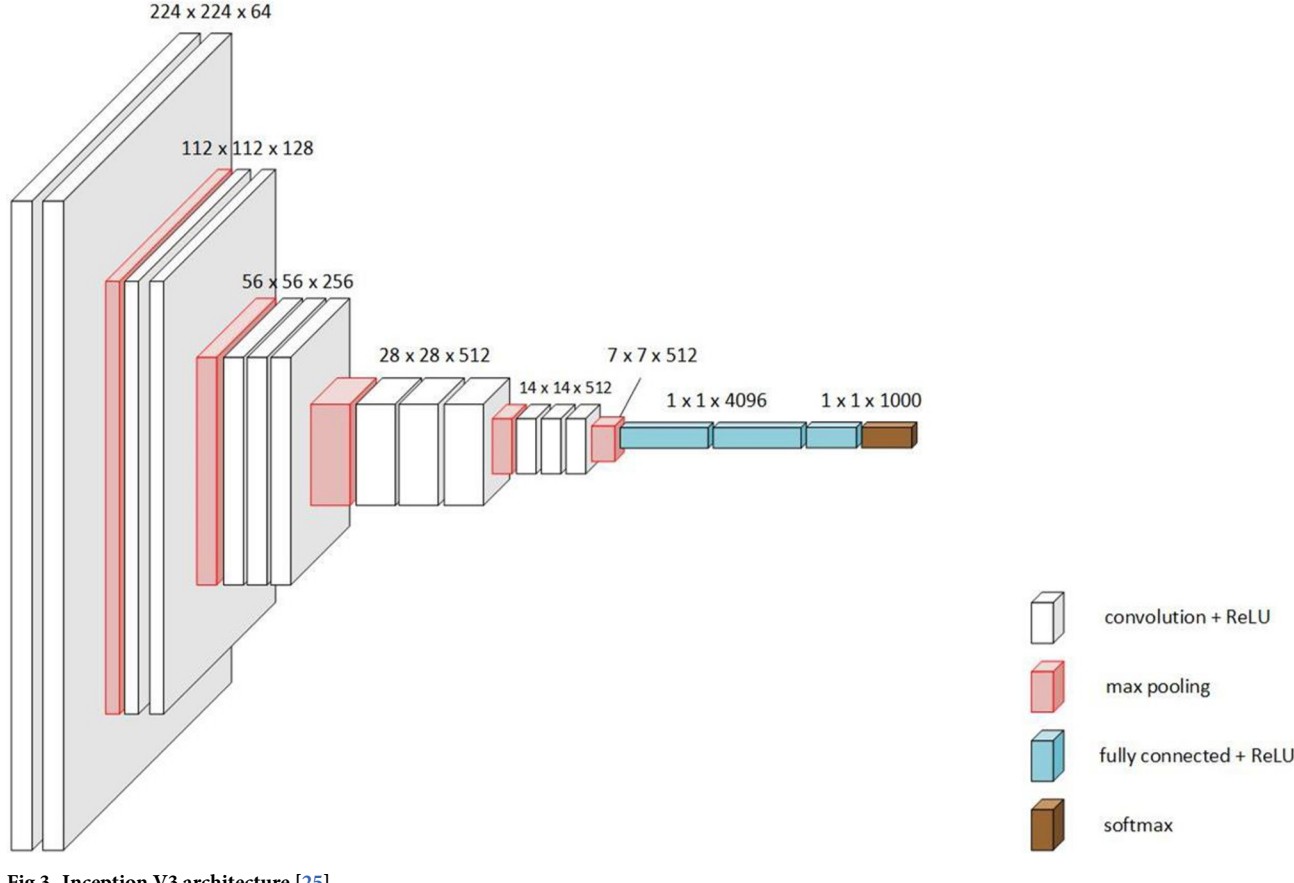

**Fig 3. Inception V3 architecture** [25].

**Xception:** Xception has 71 hidden layers and 23 million parameters. Xception was heavily inspired by Inception-v3, albeit it replaced convolutional blocks with depth-wise separable convolutions [29]. The architecture is given in Fig 5.

**ResNet-50:** At 50 layers deep and featuring 25.5 million parameters, ResNet-50 [27] was pretrained on more than a million images from the ImageNet dataset. The diagram of the model is given in Fig 6.

The comparison of key performance parameters for CNN architecture has been shown in Table 1.

**Ethical Statement:** Confirming that all experiments/analysis were performed in accordance with relevant guidelines and regulations.

## E. Data Preprocessing and augmentation

The following subsections describe the steps we used in running our models for detecting COVID-19 using 2 types of images (CT scan and Chest Xray images). We used a similar process using InceptionV3, ResNet-50, VGG19 and Xception.

**Data Splitting:** The dataset we have used is split into two sets, namely Train and Test. However, we should keep in mind that medical images are comparatively more diverse and subjective and so similar cases are observed in our dataset. It is expected that the deployment of the algorithm may not initially match with other images of real-world scenarios and thus we can receive images drastically different from those used in the training. Our dataset is split in the

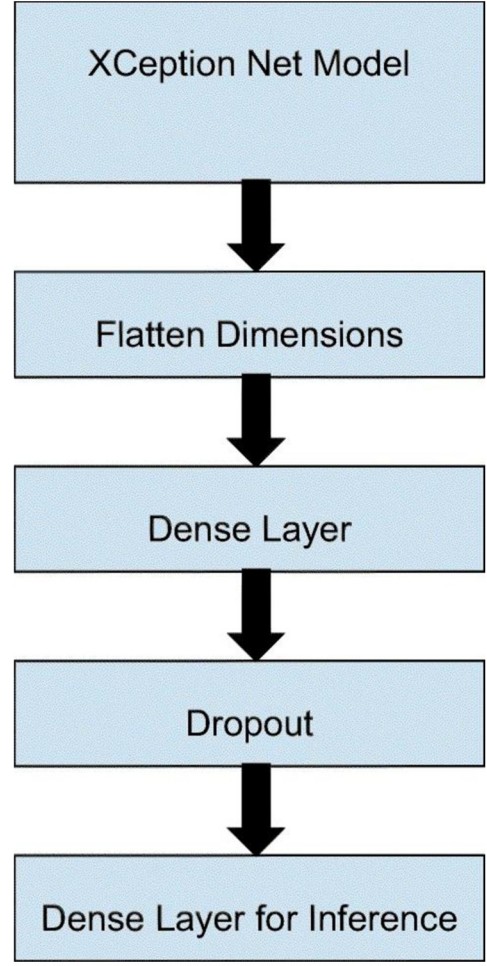

**Fig 4. VGG19 architecture [28].**

ratio of 80:20 for training and testing sets. This was randomly sampled from the dataset into the subsets.

**Data Augmentation:** Data Augmentation is the enhancing process of data points to increase data points by adding slightly modified copies of already existing data. It is commonly used in most datasets to improve classification performance [30]. Data augmentation is necessary when training data becomes complex because of shortage of samples. In our case, as the dataset contains only 700 images, we can use augmentation techniques to increase the data points. We have used data augmentation to improve the performance of our dataset. As our dataset contains almost equal numbers of images for the different classes, it is a balanced dataset. We augmented all the training data. In our case, Data Augmentation includes horizontal and vertical flips, rotation, and normalization. The techniques we have used and the values of change in data augmentation are given in Table 2. All the augmentations were applied randomly with a probability of 0.2 on a per-batch basis. Additionally, all the images were resized into 224x224.

The newly created images can then be used to train the given neural network in order to improve the training process efficiency. Besides, Augmentation can overcome the problem of overfitting and enhance the accuracy of the proposed model.

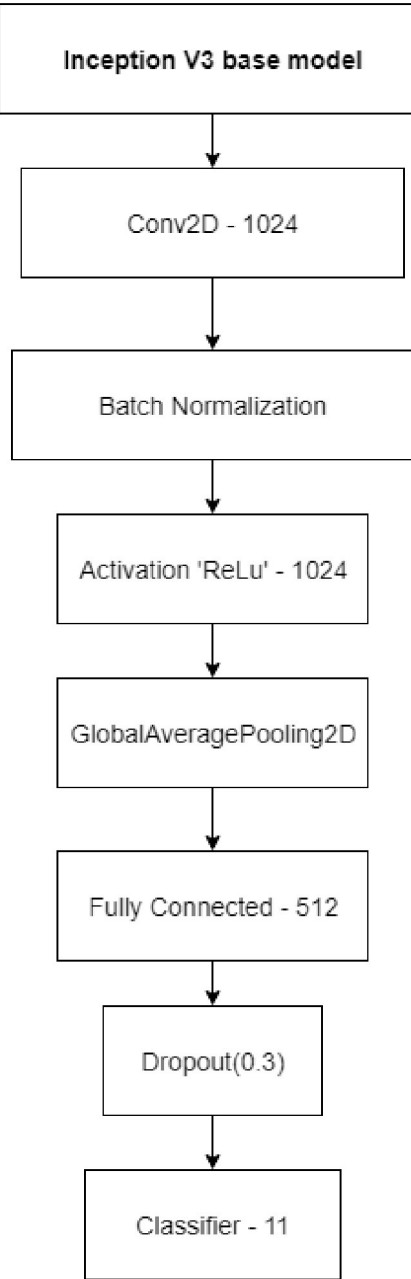

**Fig 5. Xception architecture [29].**

i. **Data Preprocessing**

## F. Experimental section

In this study, we performed deep learning analysis on 4 different datasets. Among them two of them are CT scan and X-ray images of COVID/Non-COVID patients and other two are X-ray images of bacterial pneumonia and viral pneumonia images of normal patients. Here we deployed 4 prominent deep learning models, which are VGG-19, ResNet-50, Inception V3 and Xception. In the following section we will discuss 4 test cases highlighting key

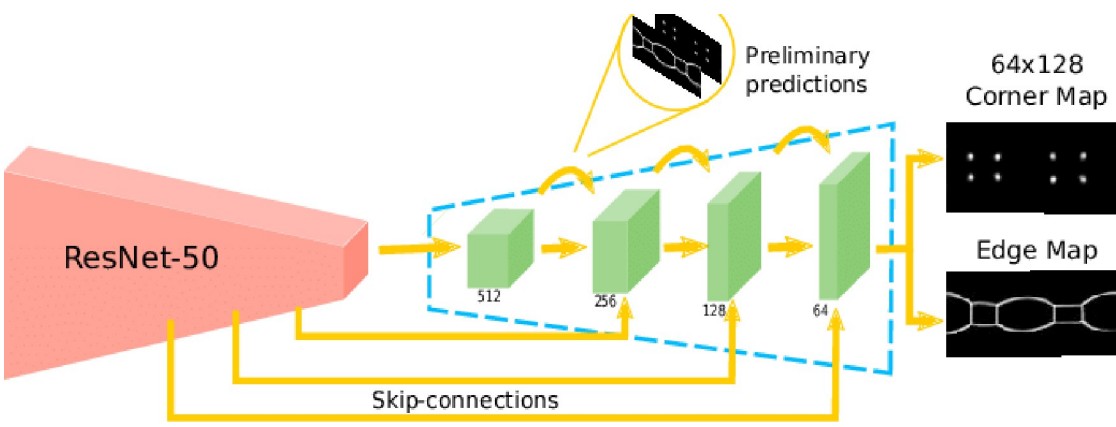

**Fig 6. ResNet-50 architecture view [27].**

performance metrics such as confusion matrix, ROC curve, model accuracy, model loss and try to find out which model is best suited for COVID detection or normal pneumonia detection.

For each model, we used a pretrained model and added fine-tuning layers. First, we used a flatten layer to flatten the input features, and then a dropout layer to reduce overfitting. After that, we used a dense fully connected layer and a softmax output to get the final prediction. For hyperparameters, we used a cross-entropy loss and an Adam optimizer.

## Case 1: CT scan Image Analysis for COVID Identification

We trained the models for 500 epochs and predicted the trained models on the test set. Then we plotted the confusion matrices, ROC curves, Classification reports, Accuracy and Loss curves. Firstly, let us look at the results using the CT scan images of COVID-19 and non-COVID-19 infected people.

i. **Confusion matrix**

A confusion matrix describes the performance of a classification model. In this chapter the True Positives are the patients who have COVID19 and are detected correctly by the algorithm. The rows in a confusion matrix correspond to what the machine learning algorithm has predicted. In our case, one will be that the patient has COVID-19 and, the other a non-COVID-19 response. The top left corner contains true positives. These are patients who have COVID-19 and are detected correctly by the algorithm. And the bottom right corner has the 'True Negatives'. These are patients who did not have COVID-19 and the algorithm correctly

**Table 1. Performance comparison among the CNN architectures.**

| Architecture Name | Year | Leading Exploit | Parameters | Depth | Strength | Challenges |
|---|---|---|---|---|---|---|
| Inception v3 | 2015 | Filter replacing capacity of large to small | 23.6 M | 159 | Minimize the computational cost by utilizing asymmetric filters and bottleneck layer | Complex architecture design with shortage of homogeneity |
| VGG19 | 2014 | Homogeneous topology with small kernel | 138 M | 19 | Capacious field with more effectiveness | Fully connected layer computation is expensive |
| Xception | 2017 | Depth wise convolution enabling | 23 M | 126 | Introduces learning across 2D followed by 3D | Increase in space and time complexity |
| ResNet-50 | 2016 | Residual learning and identity mapping | 25.5 M | 152 | Decrease error rate for deeper network | A little complex architecture |

**Table 2. Augmentation types and parameters.**

| Augmentation Type | Parameters |
|---|---|
| rotation range | 20 (rotates by 20 degrees) |
| Width_shift_range | 0.2 (take the percentage of total width as range) |
| height_shift_range | 0.2 (take the percentage of total height as range) |
| Horizontal flip | true (flips image) |

identified them for not having COVID-19. The left-hand corner contains the 'False negatives'. False Negatives are when a patient has COVID-19, but the algorithm says they don't. And the top right corner has the False positives. The false positives are patients that do not have COVID-19 but the algorithm says they do.

We took 152 samples for testing. Among them, our trained model could detect 67 of the affected patients correctly. 59 unaffected patients were also detected correctly by our model. Our model misclassified 3 patients having COVID-19 as unaffected. Only 23 patients who were uninfected were identified as infected by our algorithm as false positives. This can be seen in Fig 7.

ii. **ROC curve**

A ROC curve or receiver operating characteristic curve is a graph that shows the efficiency of a classification model at all classification thresholds. In Fig 8, we can see the combined ROC curves plotted in one plot and their AUCs.

Along the X axis of ROC curve, False Positive Rate or FPR is represented and along Y axis True Positive Rate or TPR is represented. The ROC curve shows the trade-off between sensitivity (or TPR) and specificity (1 –FPR). Classifiers that give curves closer to the top-left corner

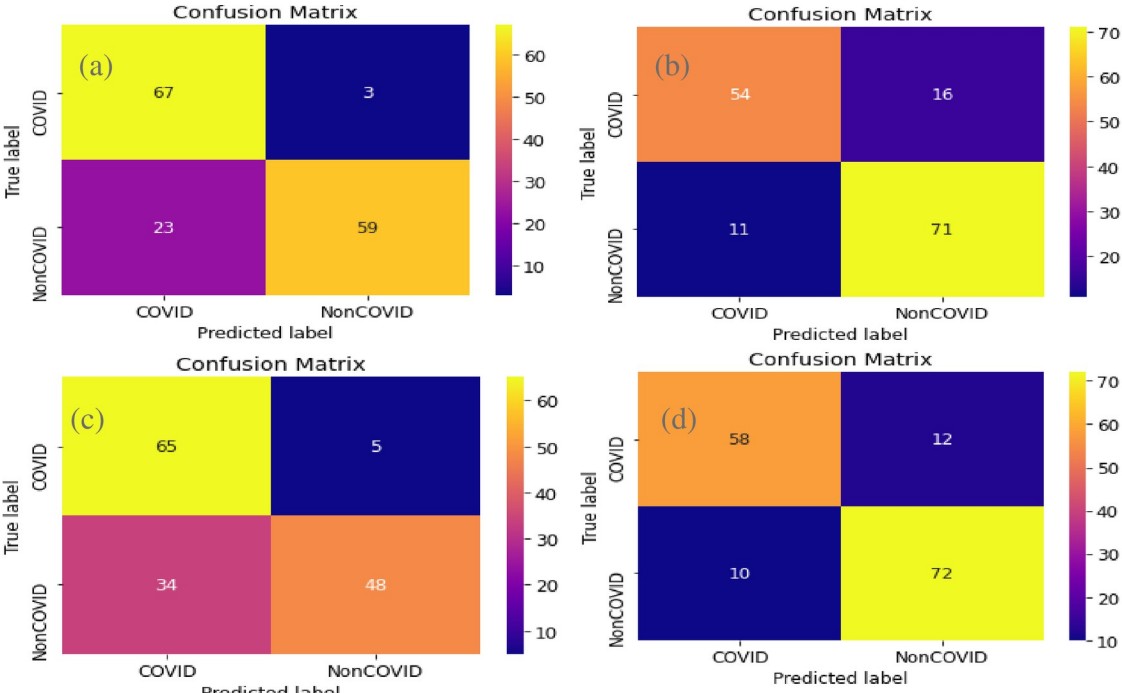

**Fig 7. Comparison of confusion matrix without normalization.** Confusion matrix for InceptionV3 (top left), VGG19 (top right), ResNet-50 (bottom left) and Xception (bottom right).

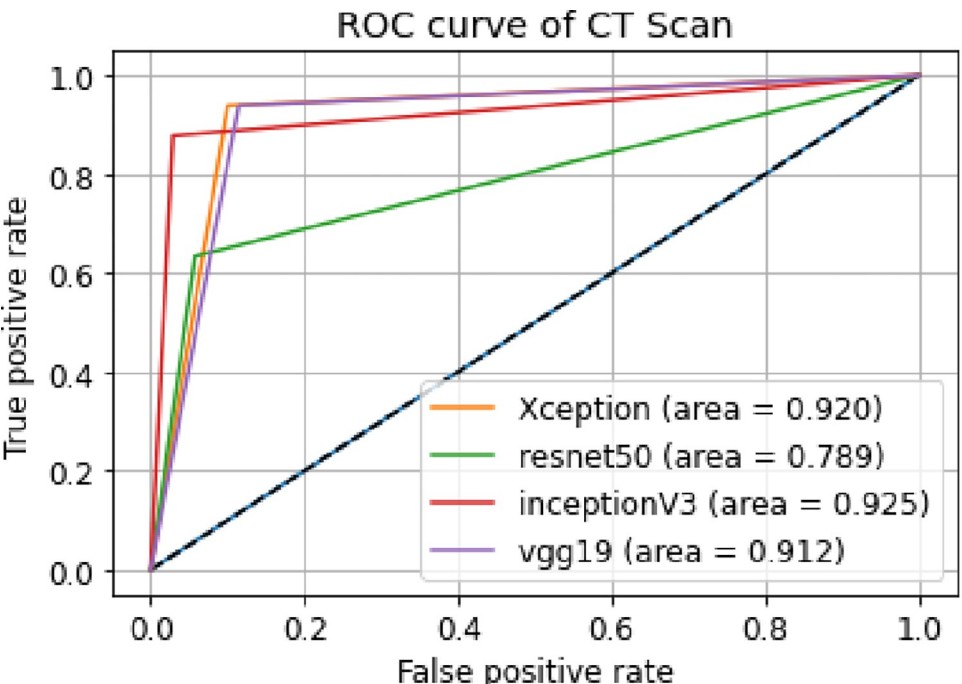

**Fig 8. Combined ROC curves of Covid CT scan.**

indicate a better performance. As a baseline, a random classifier is expected to give points lying along the diagonal (FPR = TPR). The closer the curve comes to the 45-degree diagonal of the ROC space, the less accurate the test. In the above ROC curve of the Xception model, we can see that the curve is very far away from the 45-degree diagonal of the ROC space which indicates that the curve is very accurate. The AUC also confirms our model is very accurate.

### iii. Model Accuracy

Accuracy is the number of correct predictions. We used both test and train data and calculated the accuracy after each epoch to form the accuracy curve. This is shown in Fig 9. The blue line describes the training accuracy, and the yellow line describes the test accuracy.

In the model accuracy curves. The test curve follows the train curve which happens in all cases therefore showing us that the model is working properly. The fluctuation in the line is due to the limited amount of data we had. However, we can see that the test curve is closer to the train curve in the Xception and VGG19 model.

### iv. Computing Losses

Loss is the penalty for a bad prediction. That is, loss is a number indicating how bad the model's prediction was on a single example. If the model's prediction is perfect, the loss is zero; otherwise, the loss is greater. In the model loss curves, we can find more about how the testing and training process is taking place.

In Fig 10, we can see both the Training loss and the Test loss curve is going down. In the end there is an unrepresentative split between train and test data. The curve is jumping up and down because we have limited data. The test curve should follow the train curve which happens in all cases therefore showing us that the model is working properly. Although there are some spikes in the curve, it is not to be considered and if we find the mean of the curve, we can see that the test curve is closer to the train curve in the Inception and Xception model.

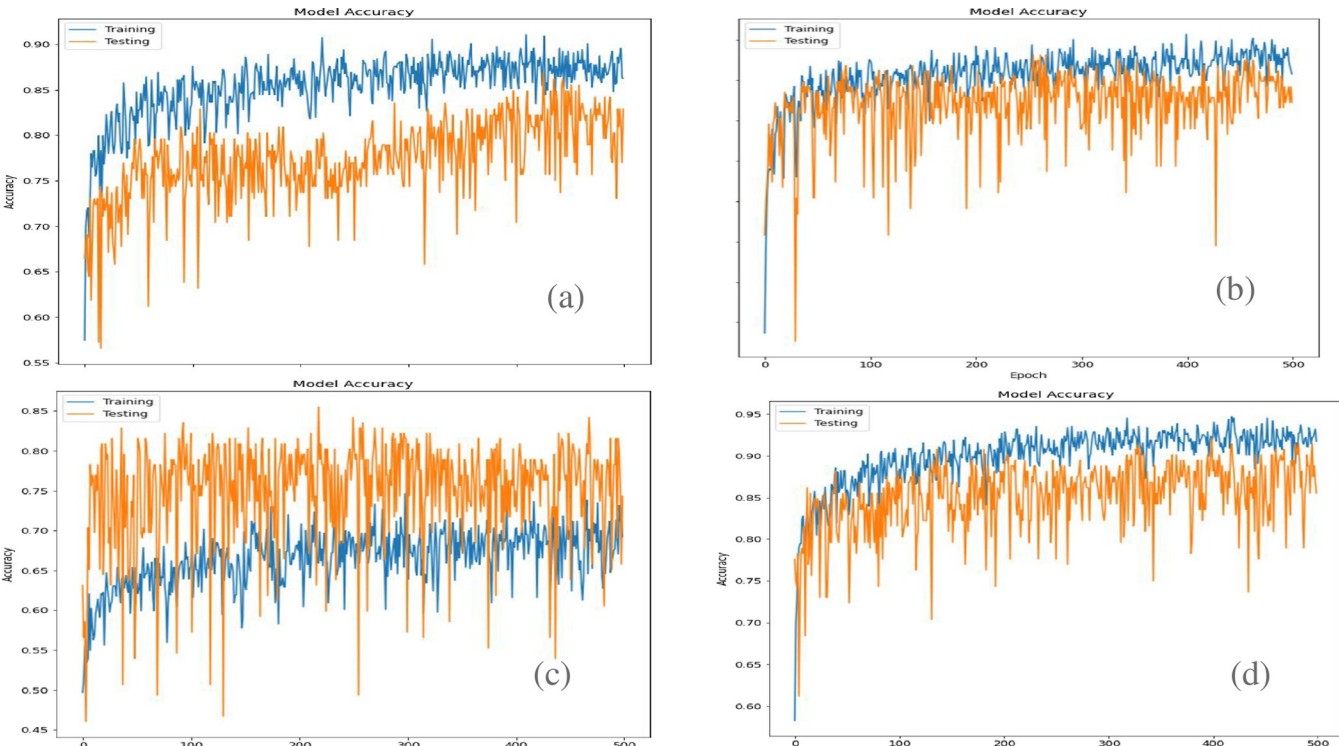

**Fig 9. Comparison of accuracy curve.** Model accuracy curve for InceptionV3 (top left), VGG19 (top right), ResNet-50 (bottom left) and Xception (bottom right).

v. **Results comparison of 4 models for CT scan images.**

From Table 3, we can get to know about some of the metrics of four models we have tested on our dataset. Among the 4 models we used, the model InceptionV3 and Xception gave us the best results, and ResNet-50 gave the least accurate results. All four models with respect to different Evaluation metrics are compared below.

## Case 2: Chest X-ray Image Analysis for COVID Identification

For Chest X-rays using Covid-19 images, we trained the models for 100 epochs and predicted the trained models on the test set. Then, we ran our custom models using Chest X-ray images of Covid patients and normal people and predicted the trained models on the test set. In this section we will see the results.

i. **Confusion matrix**

We already learnt what True Positives, True Negatives, False Positives and False Negatives are. The 'True Positives' patients have Covid and are detected correctly by the algorithm, the 'True Negatives' patients did not have Covid, and the algorithm correctly identified them for not having Covid. The 'False negatives' patient has Covid, but the algorithm says they don't. And the 'False positives' patients do not have Covid, but the algorithm says they do. Fig 11 shows the confusion matrices of the different models.

If we interpret the confusion matrices in the same way as we did before, we can see that all 4 models give really good results. VGG19 in Fig 11 has the best confusion matrix as it has 0 False Positives and False Negatives and 100% True Positive and True Negatives. That means using

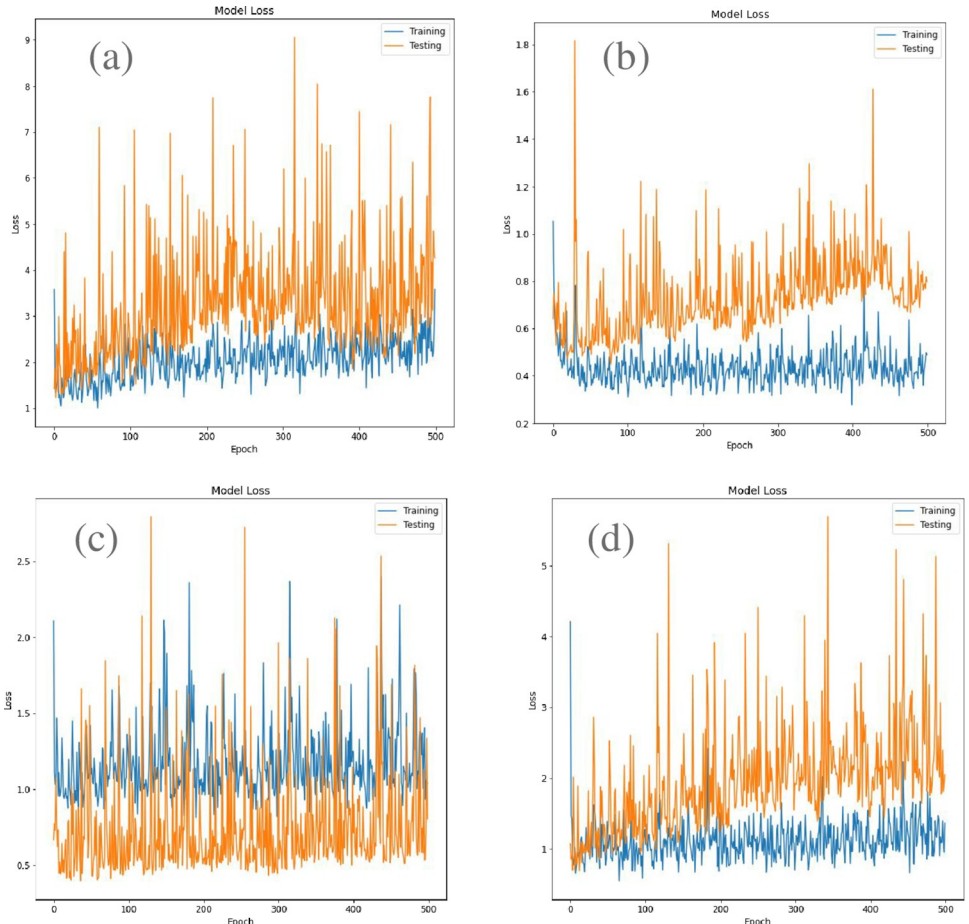

**Fig 10. Comparison of loss curves.** Model loss curve for InceptionV3 (top left), VGG19 (top right), ResNet-50 (bottom left) and Xception (bottom right).

the VGG19 model none of the results predicted were wrong. And among the samples in the test set, the trained model could detect 190 of the affected patients correctly. The 180 unaffected patients were also detected correctly by our model.

ii. **ROC curve**

A ROC curve displays the performance of a classification model at all classification thresholds. Fig 12 shows the ROC curves of our models using X-ray images dataset.

Classifiers that give curves closer to the top-left corner indicate a better performance. The Area Under Curve or AUC is used as a predictive accuracy measure for ROC curves. We can

**Table 3. Comparison of performance of different models.**

| Model | Overall Accuracy | Weighted Average | | | | Training Time per epoch (seconds) |
| --- | --- | --- | --- | --- | --- | --- |
| | | Precision | Sensitivity | F1 Score | Specificity | |
| ResNet-50 | 0.74 | 0.79 | 0.59 | 0.74 | 0.93 | 5 |
| InceptionV3 | 0.83 | 0.86 | 0.72 | 0.83 | 0.96 | 5 |
| Vgg19 | 0.82 | 0.82 | 0.87 | 0.82 | 0.77 | 7 |
| Xception | 0.86 | 0.86 | 0.88 | 0.86 | 0.83 | 5 |

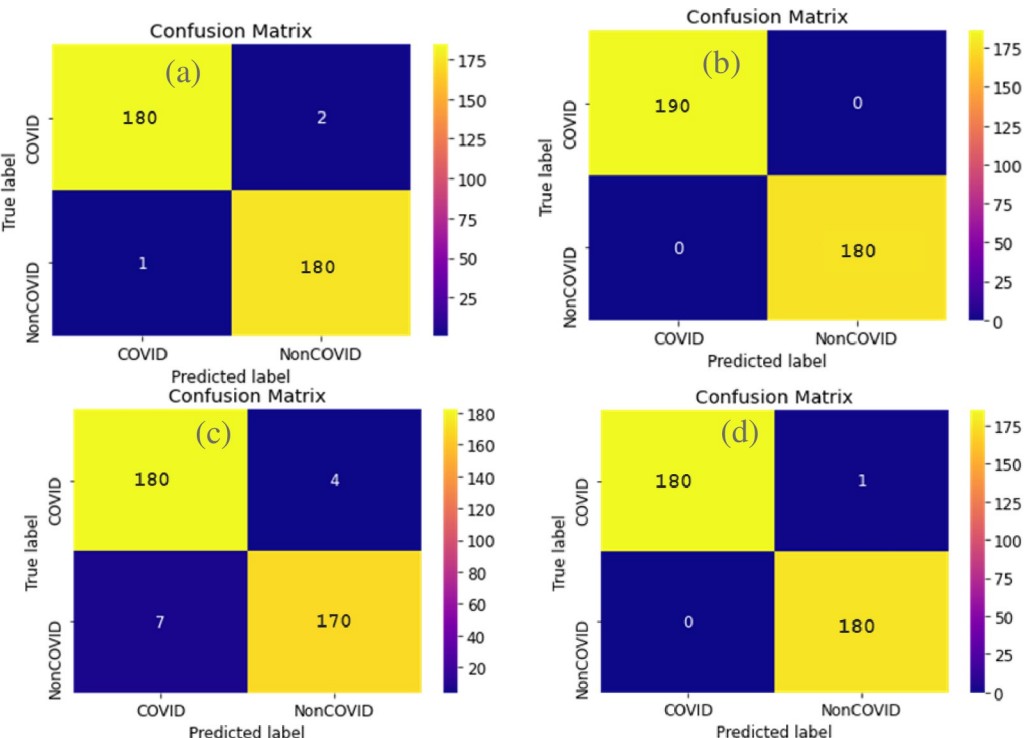

**Fig 11. Comparison of confusion matrix without normalization.** Confusion Matrix of InceptionV3 (top left), VGG19 (top right), ResNet-50(bottom left) and Xception(bottom right).

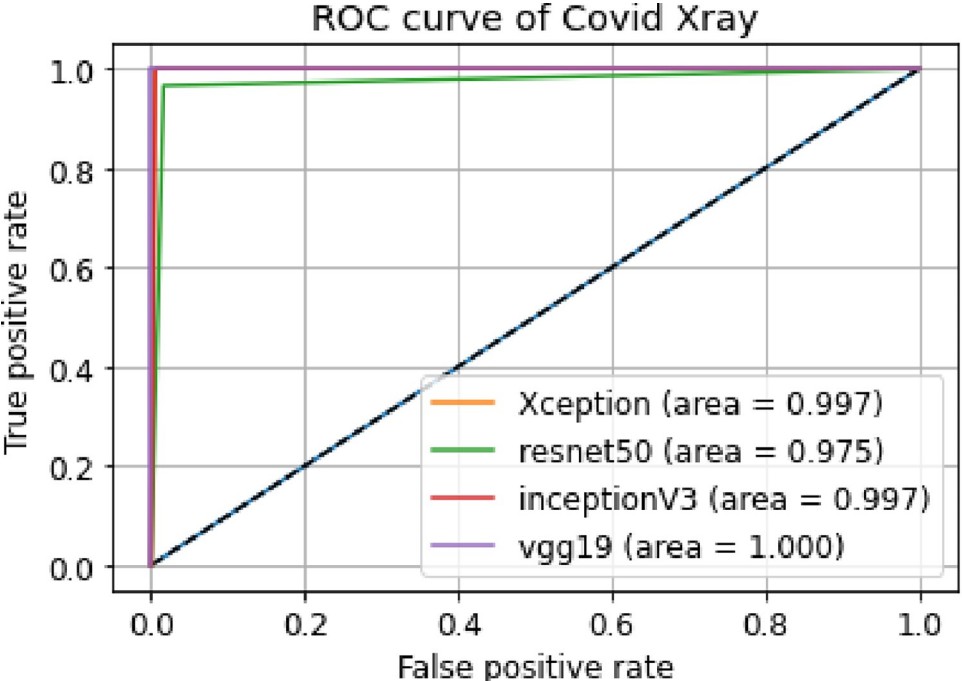

**Fig 12. Combined ROC curves of Covid Xray images.**

Table 4. Comparison of performance of different models.

| Model | Overall Accuracy | Weighted Average | | | | Training Time per epoch seconds |
|---|---|---|---|---|---|---|
| | | Precision | Sensitivity | F1 Score | Specificity | |
| ResNet-50 | 0.97 | 0.97 | 0.96 | 0.97 | 0.98 | 15 |
| InceptionV3 | 0.99 | 0.99 | 0.99 | 0.99 | 0.99 | 15 |
| Vgg19 | 1.00 | 1.00 | 1.00 | 1.00 | 100 | 17 |
| Xception | 1.00 | 1.00 | 1.00 | 1.00 | 1.00 | 15 |

see that the Area under the Curve is quite high in all 4 models. This indicates that all our 4 models are giving accurate results. The VGG19 model has the highest Area under curve as the True positive rate starts at almost 1 for false positive rate of 0.01. From Table 3 and also the ROC curves in Fig 12, we can see that the ResNet-50 model has low training time and also has the lowest accuracy in our case as the AUC is the least among the 4 models.

iii. **Model Accuracy**

We also plotted the Accuracy vs Epoch curve and found the test curve follows the train curve which happens in all cases therefore showing us that the model is working properly. Although there are some spikes in the curve, we found that InceptionV3 curves are much closer with higher accuracy. The spikes are likely due to having a relatively small dataset.

iv. **Model Loss Comparison**

We also examined the model loss vs Epoch and found both the Training loss and the Test losses are gradually decreasing. The fluctuations in the curve are because we have limited data. In the long term, the loss decreases over time, so the model is clearly learning. The test curve should follow the train curve which happens in all cases therefore showing us that the model is working properly. We found that the test curve is closest to the train curve in the InceptionV3 and Xception model.

vi. **Comparison between the Results of Covid19 using X-ray images**

From the given Table 4 we can get to know about some of the metrics of four models we have tested on our dataset. Among the 4 models we used, the model VGG19 and Xception are giving us the best metrics. However, Inception and ResNet-50 are not far behind. All four models with respect to different Evaluation metrics are compared below.

## Case 3: Chest Xray Image Analysis for Bacterial Pneumonia Identification

Xray images can also be of Bacterial Pneumonia. We trained the models for 100 epochs and predicted the trained models on the test set. In this section we will see the results using the Chest X-ray images for Bacterial Pneumonia.

i. **Confusion matrix**

A confusion matrix describes the performance of a classification model.
Fig 13 shows the confusion matrices of the different models without normalization. From the above matrices we can conclude that Xception gives the best results as the True Positives and True Negatives are close to 1 and False positive and False Negative are close to 0.

ii. **ROC curve**

Fig 14 shows the ROC curves of our models using X-ray images dataset of Bacterial Pneumonia and normal patients.

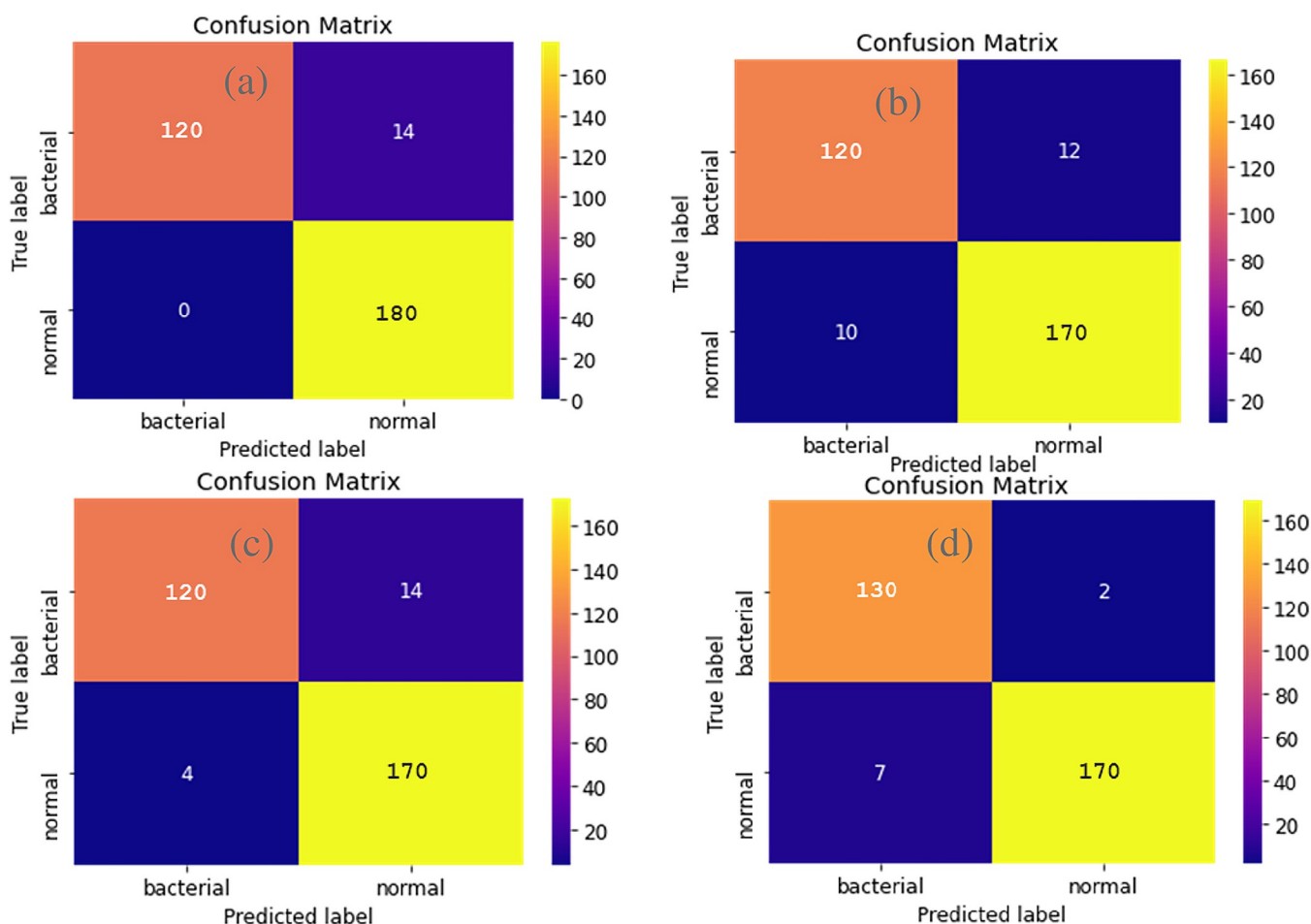

**Fig 13. Comparison of confusion matrix without normalization.** Confusion Matrix of InceptionV3 (top left), VGG19 (top right), ResNet-50 (bottom left), and Xception (bottom right).

A better output is indicated by classifiers that offer curves closer to the top-left corner. From Fig 14 we can see Xception has the highest AUC. This proves that among the 4 models, Xception gave us the best accuracy. And ResNet-50 has the least accuracy among the 4 models.

iii. **Model Accuracy**

For this data set we also examined model accuracy and found the test curve follows the train curve. Though there are some spikes and we found curves that are closer together in ResNet-50.

iv. **Model Loss Comparison**

For this dataset we found the Training loss is going down and the Test loss curve is also going down. Despite the fact it has slight ups and downs, in the long term, the loss decreases over time, so the model is learning.

The test curve should follow the train curve which happens in all cases therefore showing us that the model is working properly.

v. **Comparison between the Results of Bacterial Pneumonia for X-ray images**

From the given Table 5, we can get to know about some of the metrics of four models we have tested on our dataset. Among the 4 models we used, the model VGG19 and Xception are

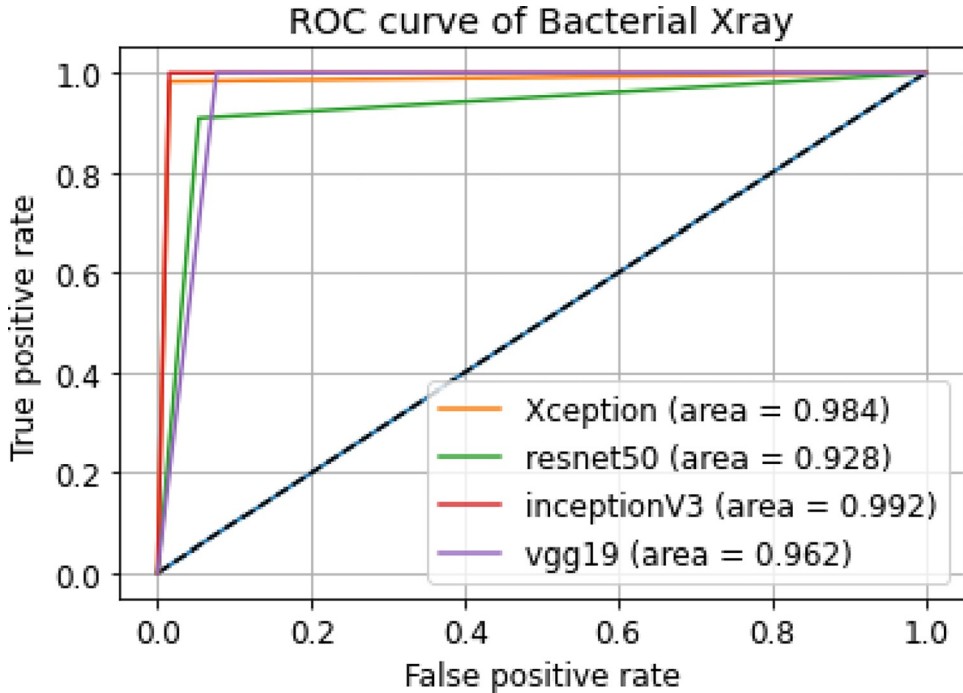

**Fig 14. Combined ROC curves of bacterial X-ray images.**

giving us the best metrics. However, Inception and ResNet-50 are not far behind. All four models with respect to different Evaluation metrics are compared below.

## Case 4: Chest X-ray Image Analysis for Viral Pneumonia Identification

We then used Viral Pneumonia Chest X-ray images since Xray images can also be of Viral Pneumonia. We trained the models for 100 epochs and predicted the trained models on the test set. In this section we will see the results using the Chest X-ray images for Viral Pneumonia.

i. **Confusion matrix**

Fig 15 shows the confusion matrices of the different models without normalization.

If we interpret the other matrices in the same way as we did before, we can see that all 4 models give really good results. VGG19 and Xception have the best confusion matrix as they have 0 False Positives and 0.096% False Negatives and 100% True Positive and 90% True Negatives.

**Table 5. Comparison of performance of different models.**

| Model | Overall Accuracy | Weighted Average | | | | Training Time per epoch seconds |
|---|---|---|---|---|---|---|
| | | Precision | Sensitivity | F1 Score | Specificity | |
| ResNet-50 | 0.93 | 0.93 | 0.94 | 0.93 | 0.91 | 14 |
| InceptionV3 | 0.94 | 0.94 | 0.98 | 0.94 | 0.89 | 13 |
| Vgg19 | 0.95 | 0.96 | 1.00 | 0.95 | 0.89 | 15 |
| Xception | 0.97 | 0.97 | 0.96 | 0.97 | 0.98 | 13 |

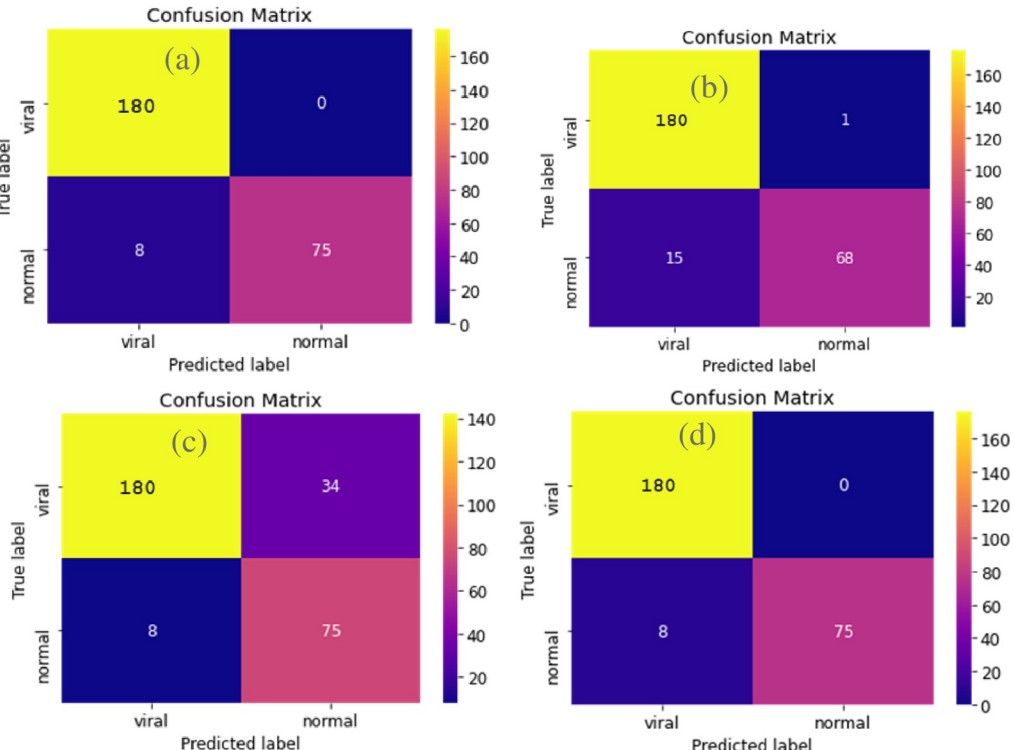

**Fig 15. Comparison of confusion matrix without normalization.** Confusion Matrix of InceptionV3 (top left), VGG19 (top right), ResNet-50 (bottom left) and Xception (bottom right).

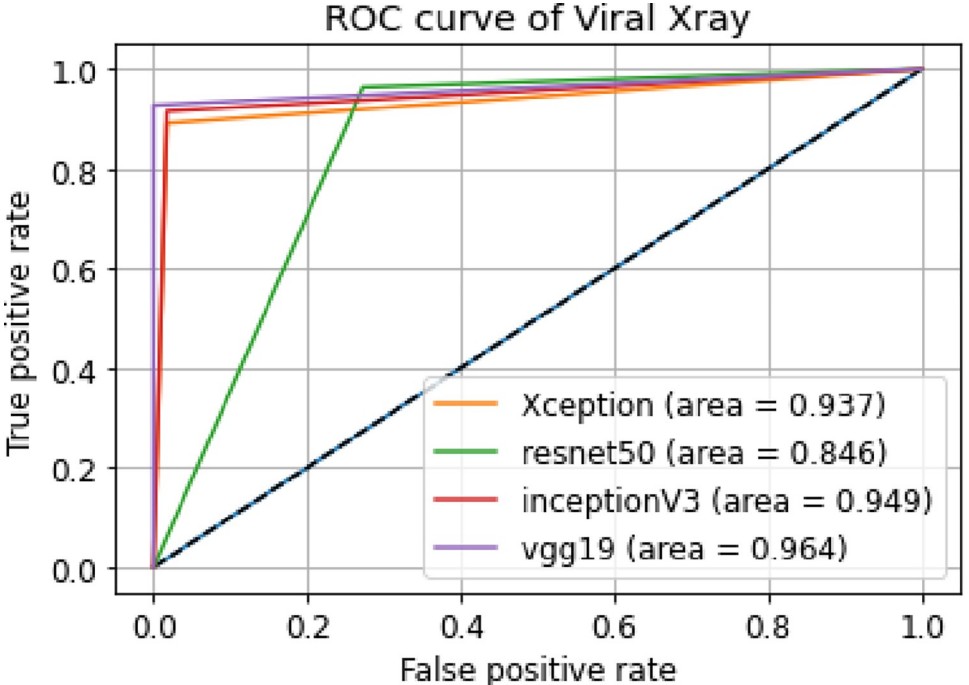

**Fig 16. Combined ROC curves of bacterial X-ray images.**

**Table 6. Comparison of performance of different models.**

| Model | Overall Accuracy | Weighted Average | | | | Training Time per epoch seconds |
|---|---|---|---|---|---|---|
| | | Precision | Sensitivity | F1 Score | Specificity | |
| ResNet-50 | 0.84 | 0.82 | 0.90 | 0.84 | 0.81 | 12 |
| InceptionV3 | 0.94 | 0.94 | 0.82 | 0.94 | 0.99 | 11 |
| Vgg19 | 0.97 | 0.97 | 0.90 | 0.97 | 1.00 | 17 |
| Xception | 0.96 | 0.96 | 0.89 | 0.96 | 0.99 | 13 |

ii. **ROC curve**

Fig 16 shows the ROC curves of our models using X-ray images dataset for Viral Pneumonia and normal patients.

We can see that the Area under the Curve is quite high in all 4 models indicating that 4 models are giving accurate results. The VGG19 and Xception models have the highest area under the curve. From the ROC curves in Fig 16, we can see that the ResNet-50 model has low training time and also has the lowest accuracy in our case as the AUC is the least among the 4 models.

iii. **Model Accuracy**

For this dataset we found the test curve follows the train curve which happens in all cases therefore showing us that the model is working properly. Although there are some spikes in the curve, it is not to be considered and if we find the mean of the curve. The curves are closest together in VGG19.

iv. **Model Loss Comparison**

In this analysis the test curve should follow the train curve which happens in all cases therefore showing us that the model is working properly. We can see that the test curve is closest to the train curve in the VGG19 and Xception model.

v. **Comparison between the results of Viral Pneumonia for X-ray images**

From the given Table 6, we can get to know about some of the metrics of four models we have tested on our dataset. Moreover, in Fig 16, we can see the combined ROC curves plotted in one plot and their AUCs. Among the 4 models we used, the model VGG19 and Xception are giving us the best metrics. However, Inception and ResNet-50 are not far behind. All four models with respect to different evaluation metrics are compared below.

Overall, on X-rays, our models perform better than some previously built models such as in [6,8,17] and are on par with others. On CT scans, our results are also close to prior work such as in [16]. Therefore, our results serve as validation for previous work as well as to establish that the same CNN architectures can perform well across both X-ray and CT images.

## G. Web deployment

To demonstrate how our models may be deployed in real-world scenarios, we developed an app using Flask [31]. The app receives image inputs from an user, and then our model sends predictions back to it. Fig 17 shows how the webpage looks when predicting.

## H. Conclusion

Diseases are part of living beings. Every animal in the world will be affected by diseases at some point of their life. Diseases can either be deadly or not. But without proper care and

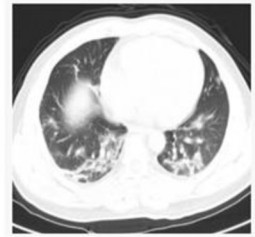 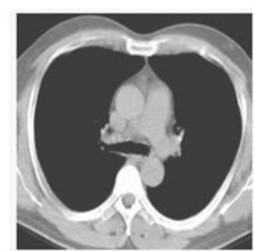

**Fig 17. COVID 19 detecting using the Flask app.**

treatment small diseases can turn up to be life threatening diseases too. In recent times humans are experiencing several deadly diseases such as cancer, malaria, tuberculosis, hepatitis and so on. However the deadliest of recent times are COVID-19 due to its super spreading nature. However, the detection of this disease for mass people is very expensive and time consuming. Many parts of the world (especially densely populated poor countries) face lots of challenges due to insufficient number of testing kits and a shortage of trained professionals.

To overcome this, using computer vision.to detect COVID-19 from X-ray and CT scan images is effective. In this study we have worked on not only to identify COVID-19 utilizing various CNN models, but also introduced a Flask app to make the detection system automated and remotely accessible via web deployment. The findings of our study reveal that among four models (ResNet-50, Inception_v3, VGG19 and Xception), Xception provides the best overall accuracy for identifying COVID patients. Additionally, we have deployed four models on two more additional dataset of X-ray images of bacterial pneumonia and viral pneumonia images of normal patients. The findings of our study show that VGG19 and Xception on X-ray images give us the best results for identifying pneumonia from viral/bacterial pneumonia.

## Author Contributions

**Conceptualization:** Nahid Islam, Abu S. M. Mohsin, Tazwar Prodhan Shaer, Md. Adnan Islam, Omar Sadat, Nahid Hossain Taz.

**Data curation:** Nahid Islam, Abu S. M. Mohsin, Shadab Hafiz Choudhury, Tazwar Prodhan Shaer, Md. Adnan Islam, Omar Sadat, Nahid Hossain Taz.

**Formal analysis:** Nahid Islam, Abu S. M. Mohsin, Shadab Hafiz Choudhury, Tazwar Prodhan Shaer, Md. Adnan Islam, Omar Sadat, Nahid Hossain Taz.

**Investigation:** Nahid Islam, Abu S. M. Mohsin, Shadab Hafiz Choudhury, Tazwar Prodhan Shaer, Md. Adnan Islam, Omar Sadat, Nahid Hossain Taz.

**Resources:** Nahid Islam.

**Software:** Nahid Islam.

**Supervision:** Abu S. M. Mohsin.

**Validation:** Abu S. M. Mohsin, Shadab Hafiz Choudhury.

**Visualization:** Nahid Islam, Abu S. M. Mohsin, Shadab Hafiz Choudhury.

**Writing – original draft:** Nahid Islam, Abu S. M. Mohsin.

**Writing – review & editing:** Abu S. M. Mohsin, Shadab Hafiz Choudhury.

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
