## [Decision Letter · Decision Letter 0]

15 Aug 2023

PONE-D-23-18788COVID-19 Detection and Web Deployment from CT scan and Xray Images Using Deep LearningPLOS ONE

Dear Dr. Mohsin,

Thank you for submitting your manuscript to PLOS ONE. After careful consideration, we feel that it has merit but does not fully meet PLOS ONE’s publication criteria as it currently stands. Therefore, we invite you to submit a revised version of the manuscript that addresses the points raised during the review process.

We look forward to receiving your revised manuscript.

Kind regards,

Maleika Heenaye- Mamode Khan

Academic Editor

PLOS ONE

Reviewers' comments:

Reviewer's Responses to Questions

**Comments to the Author**

1. Is the manuscript technically sound, and do the data support the conclusions?

Reviewer #1: Yes

Reviewer #2: Partly

2. Has the statistical analysis been performed appropriately and rigorously? 

Reviewer #1: Yes

Reviewer #2: I Don't Know

3. Have the authors made all data underlying the findings in their manuscript fully available?

Reviewer #1: Yes

Reviewer #2: Yes

4. Is the manuscript presented in an intelligible fashion and written in standard English?

Reviewer #1: No

Reviewer #2: No

5. Review Comments to the Author

Reviewer #1: COVID-19 Detection and Web Deployment from CT scan and Xray Images Using Deep Learning

Typos Errors

“Resnet50:At 50 layers deep and sporting 25.5 million parameters” : “sporting” should it not be “supporting”

Section I Confussion matrix : What is “ In this chapter”

In Figure 11, the y-axis scale is not the same for graphs (a) – (d). This can be confusing. It will be better to have the same scale for all graphs.

In Figure 12 the format for numbers in True Positive and True negative should be changed so that we can easily see the numbers.

Section H – Whole paragraph repeated:

“Our model sends predictions to the flask app. Since it is research work, we ran the service in a local machine instead of deploying online. Figure 18 below shows how to run the predictions of the model in our local environment and Figure19 shows how the webpage looks when predicting.”

Fig 18 has the same screen shot given twice.

Sections requiring additional explanations:

In the data augmentation section, elaborate on “feature wise standardization”.

“From the ROC curves in Figure 13, we can see that the ResNet50 model has low training time”, It is not clear how the curve show low training time. Please explain.

General comments:

As COVID-19 is no longer a pandemic, the work should be reframed to cover a more generalize classification of CT scan and Xray images.

Reviewer #2: The paper 'COVID-19 Detection and Web Deployment from CT Scan and Xray Images

Using Deep Learning' presents the work done and findings of the authors on the detection of covid 19 using Deep Learning. Four different DL models were trained on both x-ray images and CT-scan images.

1. Although the authors have mentioned a number of works in their literature review they have not provided the results obtained in these works and more importantly have not critically analysed existing work compared to their work. Have the authors been able to do better in their proposed solution? or are the results comparable to existing ones? or worse?

2. In the literature section, the datasets that have been selected are already mentioned and detailed – the description of the datasets should be in the experimental details/implementation

3. Reference to the figures in the text currently does not follow the Plos one formatting guidelines and is also inconsistent in the paper. In some cases, Figure 1, Figure 2 are used in the text and in some cases Fig3, Fig4, Fig 5 etc are used.

4. The Python code should not be given in the paper.

5. Table caption should be at the top of the table

6. It is unclear why the numbers in the confusion matrices are given as 1.8e+02

7. Both VGG19 and Xception have given an accuracy of 100% and no explanations were given on this. Were these cases of overfitting? Although the authors have stated that they have used a Flatten layer to flatten their input features and a Dropout layer to overcome overfitting, there seems to be overfitting here.

8. There is no need to explain how to use Flask in the paper

9. The paper is not always written in the format of a journal paper and more of a report in many cases.

10. The authors have used either 100 or 500 epochs but have not explained why they chose these numbers and what the optimum number of epochs is. Also, early stopping could have been looked into.

11. Although the users have mentioned that they have used data augmentation, they have not specified how many images they had after the augmentation step. Moreover, they did not specify whether augmentation was used with the x-ray images or for the CT-scan images.

12. Language to be improved in multiple instances. The sentences are not always clear and it makes for difficult reading of the paper.

(a) Abstract

- ‘On the contrary, current detection methods for the disease are time-consuming and expensive with low accuracy and precision.’- remove on the contrary

- ‘To address such situations, we have designed a framework for COVID-19 detection using multiple deep learning algorithms further accompanied by a deployment scheme, which can become less time consuming and highly accurate comparatively.’ reword the part after which eg which aims at reducing the time to detect the disease and errors.

- Performance achieved not mentioned in the abstract

(b) CT-scan should be used consistently in the paper, Ct-scan has also been used in some cases.

(c) D.Datasets

- CT-Scan images - ‘During the first of January [20] 2021, the dataset consisted’ should be reworded, the dataset consisted of the number of images either on the first of January or during the month of January.

- The sentence ‘The appropriateness of this dataset has been performed by a senior radiologist in Tongji Hospital, Wuhan, China, who has been assigned in diagnosis and

conducting of a bigger number of covid19 patients in the eve of the prevalence of this disease between January and April [25].’ is not clear and should be reworded.

(d) Inception v3

- The following sentence is unclear ‘Inception V3 CNN base deep neural network with 48 layers of module and can convolute 1ⅹ13ⅹ3 and 5ⅹ5 convolution.‘

- 1ⅹ13ⅹ3 and 5ⅹ5 convolution – comma to be added

- The formatting to be looked into for this section

(e) Xception

- The following sentence is unclear ‘Xception was 71 layers deep and had 23 million parameters’

(f) And many others – the English should be reviewed

6. PLOS authors have the option to publish the peer review history of their article (what does this mean?). If published, this will include your full peer review and any attached files.

Reviewer #1: **Yes: **Sunilduth Baichoo

Reviewer #2: No

---

## [Author Response · Author response to Decision Letter 0]

20 Feb 2024

Addressing Editor’s/Reviewers Feedback

Author Reply: Thank you for giving me the opportunity to submit a revised draft of my manuscript titled “COVID-19 and Pneumonia Detection and Web Deployment from CT Scan and X-ray Images Using Deep Learning ” to “PLOS ONE” journal. We appreciate the time and effort that you and the reviewers have dedicated to providing your valuable feedback on my manuscript. We are grateful to the reviewers for their insightful comments on our paper. We have been able to incorporate changes to reflect most of the suggestions provided by the reviewers. We have highlighted the changes within the manuscript in cyan/magenta color. Here is a point-by-point response to the reviewers’ comments and concerns.

Reviewer #1: COVID-19 Detection and Web Deployment from CT scan and Xray Images Using Deep Learning

Author Reply: Thank you for reviewing our paper.

Typos Errors

“Resnet50:At 50 layers deep and sporting 25.5 million parameters” : “sporting” should it not be “supporting”

Section I Confussion matrix : What is “ In this chapter”

Author Reply:We have fixed the typos and repetition issues. Thank you for noting them.

In Figure 11, the y-axis scale is not the same for graphs (a) – (d). This can be confusing. It will be better to have the same scale for all graphs.

Author Reply:Thanks for the feedback.We have fixed the issue.

In Figure 12 the format for numbers in True Positive and True negative should be changed so that we can easily see the numbers.

Author Reply:Thanks for the feedback.We have fixed the issue and numbers are visible now.

Section H – Whole paragraph repeated:

“Our model sends predictions to the flask app. Since it is research work, we ran the service in a local machine instead of deploying online. Figure 18 below shows how to run the predictions of the model in our local environment and Figure19 shows how the webpage looks when predicting.”

Author Reply:We have updated this section and taken care of repetition.

Fig 18 has the same screen shot given twice.

Author Reply:We have fixed the issue.

Sections requiring additional explanations:

In the data augmentation section, elaborate on “feature wise standardization”.

Author Reply: Feature wise standardization refers to normalization. The pixel values of the images in the dataset are normalized between 0.0 and 1.0. As feature wise standardization is a more obscure phrase, we have replaced it with the term “normalization” in Page 7.

“From the ROC curves in Figure 13, we can see that the ResNet50 model has low training time”, It is not clear how the curve show low training time. Please explain.

Author Reply:Thank you for pointing this out. There was a small omission error. The correct line is “From Table 3 and also the ROC curves in Figure 12…”. The line has been updated on Page 15.

General comments:

As COVID-19 is no longer a pandemic, the work should be reframed to cover a more generalize classification of CT scan and Xray images.

Author Reply: Thank you for noting this. The original work was carried out at a time when COVID-19 was highly relevant. We have rephrased the title and reframed the introduction to reflect this, putting a stronger focus on our focus on other forms of pneumonia, not just COVID-19, detected by CT scan and Xray images .

Reviewer #2: The paper ‘COVID-19 Detection and Web Deployment from CT Scan and Xray Images

Using Deep Learning’ presents the work done and findings of the authors on the detection of covid 19 using Deep Learning. Four different DL models were trained on both x-ray images and CT-scan images.

Author Reply:Thank you for your time and effort giving us a thorough review of the paper.

1. Although the authors have mentioned a number of works in their literature review they have not provided the results obtained in these works and more importantly have not critically analysed existing work compared to their work. Have the authors been able to do better in their proposed solution? Or are the results comparable to existing ones? Or worse?

Author Reply: We thank the reviewer for your time and effort giving us a thorough review of the paper.Our results are comparable to existing ones. We noted that existing comparative papers did not compare certain models such as Xception, and they did not holistically compare the performance of each model on X-Ray images and CT scan images together. Therefore, it was difficult to determine which type of imaging works better when using computer vision for pneumonia or COVID detection.

2. In the literature section, the datasets that have been selected are already mentioned and detailed – the description of the datasets should be in the experimental details/implementation

Author Reply: We have added a more detailed description of the way the datasets were processed in the Dataset Processing section, just before the in the experimental details section. Since the datasets are independent of the models we’re using, we believe it's important to keep the dataset description and the experimental details separate.

3. Reference to the figures in the text currently does not follow the Plos one formatting guidelines and is also inconsistent in the paper. In some cases, Figure 1, Figure 2 are used in the text and in some cases Fig3, Fig4, Fig 5 etc are used.

Author Reply: Thank you for pointing this out. We have corrected this and ensured references are consistent.

4. The Python code should not be given in the paper.

Author Reply: The python code has been removed from the paper.

5. Table caption should be at the top of the table

Author Reply: Table captions have been moved to the top of the table.

6. It is unclear why the numbers in the confusion matrices are given as 1.8e+02

Author Reply: This was an error left over from generating the original confusion matrices. This has been fixed. The confusion matrices now display all numbers in decimal format.

7. Both VGG19 and Xception have given an accuracy of 100% and no explanations were given on this. Were these cases of overfitting? Although the authors have stated that they have used a Flatten layer to flatten their input features and a Dropout layer to overcome overfitting, there seems to be overfitting here.

Author Reply: While it seems that having a 100% accuracy means overfitting, we believe that’s not the case here. This is because our training and validation loss curves are similar to each. Since the other models have accuracies over 95%, it is feasible we reached 100% accuracy while retaining generalizability.

Overfitting can occur even when test accuracy is lower than 100%. The key indicator is whether the training and validation losses and accuracies are aligned with each other or not. In this case, they are aligned, so there is no overfitting.

8. There is no need to explain how to use Flask in the paper

Author Reply: The explanation of Flask use and development has been removed.

9. The paper is not always written in the format of a journal paper and more of a report in many cases.

Author Reply: The writing of the paper has been updated. It should flow much better overall.

10. The authors have used either 100 or 500 epochs but have not explained why they chose these numbers and what the optimum number of epochs is. Also, early stopping could have been looked into.

Author Reply: We trained all the models for 500 epochs. Since we were comparing different models and image modalities, it was important to keep experimental variables constant. So, we trained all the models at an equal number of epochs and made sure it was high enough.

We did take a checkpoint when each model stopped improving and used that checkpoint for the testing.

11. Although the users have mentioned that they have used data augmentation, they have not specified how many images they had after the augmentation step. Moreover, they did not specify whether augmentation was used with the x-ray images or for the CT-scan images.

Author Reply: Data augmentation was carried out with both types of images. They were also applied randomly on a per-batch basis, so there is no fixed number of images after the augmentation steps. These details have been added to the paper.

12. Language to be improved in multiple instances. The sentences are not always clear and it makes for difficult reading of the paper.

(a) Abstract

- ‘On the contrary, current detection methods for the disease are time-consuming and expensive with low accuracy and precision.’- remove on the contrary

- ‘To address such situations, we have designed a framework for COVID-19 detection using multiple deep learning algorithms further accompanied by a deployment scheme, which can become less time consuming and highly accurate comparatively.’ Reword the part after which eg which aims at reducing the time to detect the disease and errors.

- Performance achieved not mentioned in the abstract

(b) CT-scan should be used consistently in the paper, Ct-scan has also been used in some cases.

(c) D.Datasets

- CT-Scan images – ‘During the first of January [20] 2021, the dataset consisted’ should be reworded, the dataset consisted of the number of images either on the first of January or during the month of January.

- The sentence ‘The appropriateness of this dataset has been performed by a senior radiologist in Tongji Hospital, Wuhan, China, who has been assigned in diagnosis and

conducting of a bigger number of covid19 patients in the eve of the prevalence of this disease between January and April [25].’ Is not clear and should be reworded.

(d) Inception v3

- The following sentence is unclear ‘Inception V3 CNN base deep neural network with 48 layers of module and can convolute 1ⅹ13ⅹ3 and 5ⅹ5 convolution.‘

- 1ⅹ13ⅹ3 and 5ⅹ5 convolution – comma to be added

- The formatting to be looked into for this section

€ Xception

- The following sentence is unclear ‘Xception was 71 layers deep and had 23 million parameters’

(f) And many others – the English should be reviewed

Author Reply: Thank you for the thorough feedback. The language used in the paper has been thoroughly revised. All the aforementioned unclear sentences, typos and errors have been fixed, and additional changes have been made to ensure the paper is easier to understand.

---

## [Decision Letter · Decision Letter 1]

3 Apr 2024

COVID-19 and Pneumonia Detection and Web Deployment from CT Scan and X-ray Images Using Deep Learning

PONE-D-23-18788R1

Dear Dr. Mohsin,

We’re pleased to inform you that your manuscript has been judged scientifically suitable for publication and will be formally accepted for publication once it meets all outstanding technical requirements.

Kind regards,

Maleika Heenaye- Mamode Khan

Academic Editor

PLOS ONE

Additional Editor Comments (optional):

Reviewers' comments:

Reviewer's Responses to Questions

**Comments to the Author**

1. If the authors have adequately addressed your comments raised in a previous round of review and you feel that this manuscript is now acceptable for publication, you may indicate that here to bypass the “Comments to the Author” section, enter your conflict of interest statement in the “Confidential to Editor” section, and submit your "Accept" recommendation.

Reviewer #2: All comments have been addressed

Reviewer #3: All comments have been addressed

2. Is the manuscript technically sound, and do the data support the conclusions?

Reviewer #2: Yes

Reviewer #3: Yes

3. Has the statistical analysis been performed appropriately and rigorously? 

Reviewer #2: I Don't Know

Reviewer #3: Yes

4. Have the authors made all data underlying the findings in their manuscript fully available?

Reviewer #2: Yes

Reviewer #3: Yes

5. Is the manuscript presented in an intelligible fashion and written in standard English?

Reviewer #2: Yes

Reviewer #3: Yes

6. Review Comments to the Author

Reviewer #2: The comments have been addressed, the presentation has been updated and the language has been reviewed.

Minor edit

on Page 8, i. Data Preprocessing to be removed since this section has been moved.

on Page 22, in the last paragraph of the conclusion - using computer vision.to detect COVID-19 - the '. ' to be removed between vision and to

Reviewer #3: Dear Sir,

I hope this message finds you well. I wanted to express my gratitude for your diligent efforts in addressing the feedback provided for the review of your journal paper. I have thoroughly reviewed the revised abstract and conclusion, and I'm pleased to inform you that all the comments have been effectively addressed. The revised conclusion now effectively highlights the significant outcomes of your study, emphasizing the high accuracies achieved by different deep learning models and the validation process using supplementary datasets. Moreover, the addition of insights into the implications of your findings and potential future directions adds depth and relevance to the conclusion.

Furthermore, a small suggestion would be rearding the inclusion of different ablations leading to the conclusion of using optimal hyperparameters. This addition would indeed enhance the comprehensiveness of the study and provide valuable insights into the model selection process.

Looking forward to seeing the finalized version of your paper.

7. PLOS authors have the option to publish the peer review history of their article (what does this mean?). If published, this will include your full peer review and any attached files.

Reviewer #2: No

Reviewer #3: No

---

## [Editor Report · Acceptance letter]

4 Jun 2024

PONE-D-23-18788R1 

PLOS ONE

Dear Dr. Mohsin, 

I'm pleased to inform you that your manuscript has been deemed suitable for publication in PLOS ONE. Congratulations! Your manuscript is now being handed over to our production team.

Kind regards, 

on behalf of

Dr. Maleika Heenaye- Mamode Khan 

Academic Editor

PLOS ONE